# Active Causal Learning for Conditional Average Treatment Effect Estimation

## Abstract

Estimating conditional average treatment effects (CATE) from observational data is an important problem and is of high practical relevance for many domains. Despite the great efforts of recent studies to accurately estimate CATE, most methods require complete observation of all covariates of an individual. However, in real-world scenarios, the acquisition of covariate information is usually done in a active manner, which motivates us to develop methods to minimize the total measurement cost by actively selecting the most appropriate covariates to measure while guaranteeing the CATE estimation accuracy. To this end, in this paper, we first extend the existing methods for estimating CATE to allow accurate estimation in the presence of unmeasured covariates. Next, we theoretically show the advantage of dynamically adjusting the sampling strategy based on an evolving understanding of the information measured in the covariates. Then, we formulate the dynamic sampling strategy learning as a partially observed Markov decision process (POMDP) and further develop a policy gradient method to solve the optimal dynamic policy. Extensive experiments conducted on three real-world datasets demonstrate the effectiveness of our proposed methods.

## 1 Introduction

Treatment effect estimation using observational data plays a crucial role in a broad range of domains such as precision medicine (Alaa & Van Der Schaar, 2017), digital markering (Chernozhukov et al., 2013), and policy making (Athey, 2015). For example, in healthcare, a doctor could use covariate information about a patient to estimate the conditional average treatment effect (CATE), defined as the difference between the patient's expected potential outcomes under different treatment conditions, which can be used to help determine which treatment leads to a more desired outcome. The basic challenge for accurately estimating CATE is that, since each individual can be only assigned one treatment, we always observe the corresponding potential outcome, but not both, which is also known as the fundamental problem of causal inference (Holland, 1986).

Many methods have been proposed to accurately estimate CATE. Specifically, most methods strive to balance covariates to estimate CATE accurately, such as matching, stratification, outcome regression, weighting, and doubly robust methods (Rosenbaum & Rubin, 1983; Rosenbaum, 1987; Hainmueller, 2012; Li et al., 2016). Benefiting from recent advances in deep learning, representation learning methods propose to learn a covariate representation that is independent of the treatment to overcome the covariate shift between the treatment and control groups (Johansson et al., 2016; Shalit et al., 2017b), which can be further enhanced by exploiting the local similarity presevation (Yao et al., 2018), simultaneously modeling the propensity and the potential outcomes (van der Laan & Rose, 2011; Shi et al., 2019), and disentangling the covariates Hassanpour & Greiner (2020). In addition, by exploiting the generative models, CEVAE (Louizos et al., 2017) and GANITE (Yoon et al., 2018) generate counterfactual outcomes for CATE estimation.

Despite the great efforts of recent studies to accurately estimate CATE, most methods require complete observation of all covariates of an individual. However, we believe that this is not practical in real-world scenarios–instead, the measurement of covariates should be active and well-designed. Moving back to the illustrated example in healthcare, when a doctor initially meets a new patient, it is hardly possible for the doctor to have all the covariate information about the patient. Instead, the doctor might ask the patient to take some medical tests, such as drawing blood or taking medical

images. By taking such medical tests, the doctor can collect more, but not all, of the patient's covariate information. Based on the collected covariate information, the doctor can either (1) collect more covariates by letting the patient take more medical tests or (2) stop collecting extra covariates by recognizing that the already collected covariates are sufficient for accurately estimating CATE and making the treatment decisions for that patient. In addition, it is reasonable to assume that there is a cost associated with the measurement of each covariate. Therefore, it is necessary to develop methods to minimize the total measurement cost by actively selecting the most appropriate covariates to measure while guaranteeing the CATE estimation accuracy.

To fill this gap, this paper studies the active covariate measurement for treatment effect estimation. The main contributions are summarized below:

• We extend the existing representation learning methods for estimating CATE to allow accurate estimation in the presence of unmeasured covariates. Specifically, we introduce a learnable embedding lookup table for each covariate, and design a uniform sampling approach to make the CATE estimation robust to different numbers of measured covariates as inputs.
• We consider how to select the covariates to be observed, making it possible to maximize the estimation accuracy while minimizing the associated costs. Through theoretical analysis, we demonstrate the superiority of dynamic policy over static policy, where the latter employs a fixed set of covariates for all instances, whereas the former dynamically adjusts its sampling strategy based on an evolving understanding of the information measured in the covariates.
• We formulate the dynamic sampling strategy learning as a Partially Observed Markov Decision Process (POMDP) and develop a policy gradient method to solve the optimal dynamic policy. This allows us to model the dynamic sampling process as a sequence of decisions made under uncertainty, with the aim of maximizing the expected sum of rewards over time.
• Experiments on real-world datasets show our method can effectively achieve active covariate measurement, ensuring the accuracy of CATE estimation while minimizing the measurement cost.

## 2 RELATED WORK

**Dynamic Sampling.** Dynamic sampling strategy adaptively collects measurements based on information from previous measurements. Recent works have effectively framed these challenges as Markov decision processes (MDPs) and have approached solutions using reinforcement learning (RL) (Li, 2017). Examples abound in various domains, including the Travelling Salesman Problem (Bello et al., 2016), Vehicle Routing Problem (Kool et al., 2018), and Influence Maximization (Manchanda et al., 2019). These studies have consistently demonstrated that RL-based policies can outperform static or greedy approaches in terms of efficiency and effectiveness. In our work, by formulating the active covariate measurement problem as a POMDP (Sondik, 1971; Kaelbling et al., 1998), we extend the above methods to minimize the total measurement cost by actively selecting the most appropriate covariates to measure while guaranteeing the CATE estimation accuracy.

**CATE Estimation.** Benefiting from recent advances in machine learning, many methods have been proposed for estimating CATE, including matching methods (Rosenbaum & Rubin, 1983; Schwab et al., 2018; Yao et al., 2018), tree-based methods (Chipman et al., 2010; Wager & Athey, 2018), representation learning methods (Johansson et al., 2016; Shalit et al., 2017b; Shi et al., 2019; Wu et al., 2022; Wang et al., 2023), and generative methods (Louizos et al., 2017; Yoon et al., 2018; Wu & Fukumizu, 2021). Unlike the existing work devoted to estimating CATE with complete observation of all covariates of an individual, our work focuses on a more practical setting, in which the covariates is measured from a active manner with varying costs.

## 3 PRELIMINARIES

We consider the case of binary treatment. Suppose a simple random sampling of $n$ units from a super population $\mathbb{P}$, for each unit $i$, the covariates and the assigned treatment are denoted as $X_i = (X_{i,1}, \ldots, X_{i,p}) \in \mathbb{R}^m$ and $W_i \in \{0, 1\}$, respectively, where $W_i = 1$ and $W_i = 0$ means receiving and not receiving the treatment, respectively. Let $Y_i \in \mathbb{R}$ be the outcome of interest. To study CATE estimation, we adopt the potential outcome framework (Rubin, 1974; Neyman, 1990) in causal inference. Specifically, let $Y_i(0)$ and $Y_i(1)$ be the outcome of unit $i$ had this unit receive treatment $W_i = 0$ and $W_i = 1$, respectively. Since each unit can be only assigned with one treatment,

we can only observe the corresponding outcome to be either $Y_i(0)$ or $Y_i(1)$, but not both, which is the well-known fundamental problem of causal inference (Holland, 1986; Morgan & Winship, 2015).

For unit $i$, the individual treatment effect (ITE) is defined as $\text{ITE}_i = Y_i(1) - Y_i(0)$, which indicates that whether the treatment $W_i = 1$ is beneficial for individual $i$. The conditional average treatment effect (CATE) is defined as

$$\tau(x) = \mathbb{E}[\text{ITE}_i | X_i = x] = \mathbb{E}[Y_i(1) - Y_i(0) | X_i = x], \tag{1}$$

which is the difference in the conditional mean of potential outcomes given fully measured covariates.

To identify CATE, we assume that the observation for unit $i$ is $Y_i = (1 - W_i)Y_i(0) + W_i Y_i(1)$. In other words, the observed outcome is the potential outcome corresponding to the assigned treatment, which is also known as the consistency assumption in the causal literature. We assume that the stable unit treatment value assumption (STUVA) assumption holds, *i.e.*, there should be no alternative form of treatment and interference between units. Furthermore, we require the strong ignorability assumption $(Y_i(0), Y_i(1)) \perp\!\!\!\perp W_i | X_i$ and the positivity assumption $\eta < \mathbb{P}(W_i = 1 | X_i = x) < 1 - \eta$, where $\eta$ is a constant between 0 and $1/2$.

## 4 PROPOSED METHOD

In many real-world settings (*e.g.* healthcare), the covariates should be partially observed, which leads to the first question: how to estimate the treatment effect, in the absence of some variable? In this work, we are concerned with the CATE based on an arbitrary subset of covariates. Formally, given a binary mask vector $M$, its measured index set is denoted as $\mathscr{A}(M) := \{a | M(a) = 1\} \subset \{1, 2, \ldots, p\}$, and its measured covariates set is denotes as $\mathscr{X}(X, M) := \{X_a | a \in \mathscr{A}(M)\}$. Then, we seek to estimate the CATE conditional on the partially measured covariates $\mathscr{X}(X, M)$ defined as:

$$\tau_M(\tilde{x}) = \mathbb{E}(Y_i(1) - Y_i(0) | X_{i|M} = \tilde{x}). \tag{2}$$

Furthermore, the active process of measuring covariates enlightens our second question: what order of measurements and stopping criterion would provide the best balance between measurement cost and estimation accuracy? In this work, we consider a dynamic sampling policy that can adaptively decide to either measure which covariate or stop sampling according to the partially observed covariates at each acquisition step. Formally, we want to learn a policy $\pi$, which takes $\mathscr{X}(X, M)$ as input, and predicts from $\{1, \cdots, p\}$ as the next measuring index or 0 as the stop signal. As the process continues according to the policy's guidelines, a mask vector $M$ is eventually obtained. Given a cost function $c : \{1, \cdots, p\} \to \mathbb{R}$, the policy $\pi$ aims to minimize a trade-off between the accumulative costs $\sum_{a \in \mathcal{A}(M)} c(a)$ and the estimation accuracy Equation (2).

**Methodology Overview.** In Section 4.1, we design a method for estimating causal effects in the partially absence of covariates. In Section 4.2, we consider a dynamic sampling policy that decides which covariate to observe next by the covariates that have already been observed and gives a theoretic guarantee that the optimal dynamic sampling policy is better than the static one. In Section 4.3, we formulate the dynamic sampling problem as a Partially Observed Markov Decision Process (POMDP) and solve it via a modified Proximal Policy Optimization (PPO) algorithm.

### 4.1 COUNTERFACTUAL REGRESSION WITH MASKED COVARIATES

We commence by tackling the estimation of causal effects under covariate absence. Previous treatment effect estimation methods typically rely on complete observations of covariates, which use two estimation models $f_0(X)$ and $f_1(X)$ to estimate $Y(0)$ and $Y(1)$ respectively. We extend these methods by considering a mask vector $M$ as part of the input of $f_0$ and $f_1$, which indicates that two estimation models only can make predict with measured covariates $\mathscr{X}(X, M)$.

The core of our method lies in the optimization of two these two estimation models, $f_0(\mathscr{X}(X, M))$ and $f_1(\mathscr{X}(X, M))$ designed to predict outcomes for control and treatment groups. respectively. We use TARNet for illustrative purposes, and our approach can also be used for other CATE estimation methods. Under the condition of missing covariates, the optimization objective is formulated as the prediction error between the estimated outcomes and the observed outcomes:

$$\min_{\{f_0, f_1\}} \frac{1}{|\mathcal{D}|} \sum_{(X, W, Y) \in \mathcal{D}} \|f_W(\mathscr{X}(X, M)) - Y(W)\|^2.$$

In particular, both $f_0$ and $f_1$ are inspired by the architecture of TARNet, with modifications to accommodate masked covariates. A mask vector $M$ is integrated into the data preprocessing layer $\Phi(X, M)$. It regularises continuous covariates to $[0, 1]$ and is set to $-1$ if the covariate is unmeasured. A categorical covariate (assuming $m$ classes), on the other hand, is projected onto $(m+1)$ learnable embedding vectors, where the extra one is used to indicate that the covariate is missing.

Table 1: Notation summary table.

| Terminology | Notation |
| --- | --- |
| dataset | $\mathcal{D} = \{X, W, Y\}$ |
| covariates | $X_i = (X_{i,1}, X_{i,2}, \cdots, X_{i,p})$ |
| mask vector | $M \in \{0, 1\}^p$ |
| $a$-th dimension of $M$ | $M(a) \in \{0, 1\}$ |
| measured index | $\mathscr{A}(M) := \{a \mid M(a) = 1\}$ |
| measured covariates | $\mathscr{X}(X, M) := \{X_a \mid a \in \mathscr{A}(M)\}$ |
| estimation model | $f_w : f_w(\mathscr{X}(X, M))$ to estimate $Y(w)$ |
| estimated outcome | $\hat{Y}(T) := f_w(\mathscr{X}(X, \mathbf{1}))$ |
| policy | $\pi : \pi(\mathscr{X}(X, M))$ to predict action |
| cost function | $c : \{1, 2, \cdots p\} \to \mathbb{R}$ |

Covariates that have been processed by the data preprocessing layer are concatenated together as inputs to the following two regression layers $h_0$ and $h_1$ which connect to estimate $Y(0)$ and $Y(1)$, respectively. Therefore, we denote $f := (f_0, f_1) = (h_0 \circ \Phi, h_1 \circ \Phi)$.

The sampling method for the mask vector $M$ is designed to ensure robust learning across different levels of covariate observability. In practice, we employ a uniform distribution to decide $|M|_0$ from 1 to $p$, *i.e.*, the total number of measured covariates. We then randomly generate mask vectors that conform to this pre-determined number. As opposed to simply sampling uniformly from $\{0, 1\}^p$, this strategy guarantees that our models are adept at learning from scenarios with both sparse and abundant covariate information.

## 4.2 COVARIATE MEASUREMENT POLICY EVALUATION

Given trained estimation models with masked covariates $f$, our next consideration is how to select the covariates to be observed, making it possible to maximize the estimation accuracy while minimizing the associated accumulative costs. In the real world, measurements of covariates are usually *step by step*, which leads us to get more information after each observation. Inspired by this, a superior policy $\pi$ should be adaptive: it should dynamically adjust its sampling strategy based on an evolving understanding of the information measured in the covariates.

Motivated by the above, we propose a novel dynamic sampling methodology. It commits to adaptively selecting covariates within the dynamically evolving landscape of data, optimizing the weighted accumulative costs of covariate measurement, and accuracy of CATE estimation. In particular, we develop a policy model $\pi$, which is tasked to make sequential decisions, that is, dynamic sampling of covariates over time. Formally, the goal of policy learning can be encapsulated by the following:

$$\min_{\pi} \quad \frac{1}{|\mathcal{D}|} \sum_{(X, W, Y) \in \mathcal{D}} \left[ \sum_{t=1}^{p} c(a_t) + \lambda \hat{\mathcal{L}}_f(X, M_p, W, Y) \right]$$

$$\text{s.t.} \quad a_t \sim \pi(\mathscr{X}(X, M_{t-1})), \quad t = 1, \cdots, p,$$
$$M_t = \mathbb{1}(M_{t-1} + e_{a_t}), \quad t = 1, \cdots, p, \tag{3}$$

where $c(a)$ represents the cost function for selecting the covariate at $a$-th dimension, $\lambda$ is a balancing weight parameter, $M_0$ is an all-zero vector $\mathbf{0}$. The policy satisfies that $\{0, 1, \cdots, p\} \sim \pi(\mathscr{X}(X, M))$, where $0 \sim \pi(\mathscr{X}(X, M_T))$ indicates that policy predicts to stop generation and sets $T$ as the terminal time. Furthermore, we extend $c(0) = 0$ and $e_0 = \mathbf{0}$. After the terminal time $T$, the policy will always give 0, and thus $M_t$ remains constant for $t \geq T$. The estimation error is defined as

$$\hat{\mathcal{L}}_f(X, M, W, Y) = \left\{ [f_w(\mathscr{X}(X, M)) - f_{1-w}(\mathscr{X}(X, M))] - [Y(w) - \hat{Y}(1-w)] \right\}^2.$$

In contrast to the proposed dynamic approach, another traditional sampling process is a static policy that employs a fixed set of covariates for all instances, determined a priori. The optimization problem for a static policy is defined as:

$$\min_{M} \frac{1}{|\mathcal{D}|} \sum_{(X, W, Y) \in \mathcal{D}} \left[ \sum_{a \in \mathscr{A}(M)} c(a) + \lambda \hat{\mathcal{L}}_f(X, M, W, Y) \right]. \tag{4}$$

Our approach is anchored in a theoretical foundation that highlights its superiority over conventional static methods in variable selection. The theoretical cornerstone of our dynamic sampling methodology lies in its ability to adaptively refine the covariate selection process. The dynamic policy's advantage over its static counterpart is formalized below.

**Theorem 4.1.** *Given the estimation models $f$ and weight parameter $\lambda$, the optimal value of Equation* (3) *for dynamic policy is no less than that of Equation* (4) *for static policy.*

*Proof of Theorem 4.1.* For the optimization problem Equation (4), the feasible region of the optimization objective $M$ is finite. Therefore, there must exist an optimal solution, denoted as $M^*$. We define the same pattern policy $\pi^{M^*}(\mathscr{X}(X,M)) = \mathrm{Uniform}(\mathcal{A}(M^*) \setminus \mathcal{A}(M))$ when $\mathcal{A}(M^*) \setminus \mathcal{A}(M) \neq \emptyset$, and $\pi^{M^*}(\mathscr{X}(X,M)) = 0$ otherwise. For each dataset $\mathcal{D}$, at the terminal state, we have $\pi^{M^*}$ with mask vector $M^*$. Therefore, the objective function of Equation (3) is equal to the optimal value of Equation (4) when $\pi = \pi^{M^*}$. Moreover, $\pi^{M^*}$ is also in the feasible region of the optimization problem Equation (3). Therefore, the optimal value of Equation (3) is no less than the value of the objective function of Equation (3) when $\pi = \pi^{M^*}$, as well as the optimal value of Equation (4). $\qquad\square$

This theorem claims that our dynamic sampling policy is at least as strong as a static sampling policy. This is because our dynamic method continuously adjusts the selection of covariates in response to evolving data patterns, a feature starkly missing in static approaches. However, static methods, which fix covariates based on initial data insights, may fail to capture subsequent data variations, potentially leading to suboptimal CATE estimations. Furthermore, we will demonstrate empirically that a static sampling policy is generally less effective than our dynamic sampling policy.

### 4.3 DYNAMIC COVARIATE MEASUREMENT POLICY LEARNING

Building upon the dynamic sampling optimization problems, we now turn our focus to solving the sequential decision optimization problem in Equation (3).

A sequential decision problem can be formulated as a Partially Observed Markov Decision Process (POMDP), which is a tuple $(\mathcal{S}, \mathcal{O}, \mathcal{A}, \gamma, \mathbb{P}, r)$ that consists of the state space $\mathcal{S}$, the observation space $\mathcal{O}$, the action space $\mathcal{A}$, the discount factor $\gamma$, the deterministic transition function of the environment $\mathbb{P}: \mathcal{S} \times \mathcal{O} \times \mathcal{A} \to \mathcal{S} \times \mathcal{O}$ and the reward $r: \mathcal{S} \times \mathcal{O} \times \mathcal{A} \to \mathcal{R}$. A policy $\pi$ in RL is a probability distribution on the action $\mathcal{A}$ over $\mathcal{O}$: $\pi: \mathcal{O} \times \mathcal{A} \to [0,1]$. Denote the interactions between the agent and the environment as a trajectory $\tau = (s_0, o_0, a_1, r_1, ...)$. The return of $\tau$ is the discounted sum of all its future rewards $G(\tau) := \sum_{t=1}^{\infty} \gamma^{t-1} r_t$. Given an MDP, the goal of a reinforcement learning algorithm is to find a policy $\pi$ that maximizes the discounted accumulated rewards in this MDP:

$$\max_{\pi} \mathbb{E}_{s_0 \sim \rho(s)} \mathbb{E}_{\tau}[G(\tau)|\tau(s_0) = s, \tau \sim \pi],$$

where $\rho(s)$ is an initial state distribution. In our study, we formulate the proposed dynamic sampling problem as a POMDP, where the objective is to make sequential decisions on covariate selection under uncertainty. The POMDP framework is formulated in below:

• State: $s_t$. We define the state $s_t = (X, W, Y, M_t)$, where $(X, W, Y)$ is invisible to the policy and invariant over time, and mask vector $M_t$ controls which covariates are visible.

• Observation: $o_t$. We define the observation $o_t = \mathscr{X}(X, M_t)$ as the measured covariates which is derived from the current state and represents the information available to the policy. In POMDP, at each time step, the RL agent can only observe $o_t$.

• Action: $a_{t+1}$. We define the action as the consist of the selected index of covariants and the stopping criteria at each time step. In the RL process, $a_{t+1} \in \{0, 1, \cdots, p\}$ samples from $\pi(\mathscr{X}(X, M_t))$, which is a $(n+1)$-dimensional discrete probability distribution. When $a_{t+1} \neq 0$, it indicates the selected index, otherwise it releases the signal for this process to stop.

• Transition: $\mathcal{P}$. After the action $a_t$ is chosen, the state $s_{t-1} = (X, W, Y, M_{t-1})$ transitions to $s_t = (X, W, Y, M_t)$, i.e., the mask vector $M_{t-1}$ transitions to $M_t$. We update it as:

$$M_t = \mathbb{1}(M_{t-1} + e_{a_t}), \quad \text{if} \quad 1 \leq a_t \leq p,$$

where $\mathbb{1}(\cdot)$ is the indicator function, or set current as the terminal time $T = t$, if the policy returns a null action $a_t = 0$ or the time reaches the terminal $t = p$.

---

**Algorithm 1** Dynamic Covariate Measurement with $\pi$

---

**Require:** covariates $X$, weight parameter $\lambda$ and policy $\pi$
1: Initialize $t \leftarrow 0$, $T \leftarrow p$ and $M_0 \leftarrow \mathbf{0}$;
2: **while** $t < T$ **do**
3:   Sample action $a_{t+1} \sim \pi(\mathscr{X}(X, M_t))$;
4:   Update timestep $t \leftarrow t + 1$;
5:   **if** $1 \leq a_t \leq p$ **then**
6:     Update the mask vector $M_t = \mathbb{1}(M_{t-1} + e_{a_t})$;
7:   **else**
8:     Stop the trajectory $T \leftarrow t$;
9:   **end if**
10: **end while**
**Output:** trajectory $\tau = (\mathscr{X}(X, M_0), a_1, \mathscr{X}(X, M_1), ...)$.

---

• Reward: $r_t$. The reward $r_t = r_t(s_t, a_t)$ represents how much benefit an action performed in the current state would bring in the current state. We make sure it quantifies the value of Equation (3) decreasing by select $a_t$-th covariate

$$-c(a_t) - \lambda(\hat{\mathcal{L}}_f(X, M_t, W, Y) - \lambda\hat{\mathcal{L}}_f(X, M_{t-1}, W, Y)).$$

When $a_t = 0$, the RL process stops and the agent does not need a reward.

• Discount factor: $\gamma \in [0, 1]$. It determines how much the RL agent cares about rewards in the distant future relative to those in the immediate future, which is a hyper-parameter.

This formulation allows us to model the dynamic sampling process as a sequence of decisions made under uncertainty. The goal is to develop a policy maximizing the expected sum of rewards over time:

$$-\sum_{t=1}^{T} c(a_t) - \lambda(\hat{\mathcal{L}}_f(X, M_T, W, Y) - \hat{\mathcal{L}}_f(X, M_0, W, Y)).$$

Since the last component $\hat{\mathcal{L}}_f(X, M_0, W, Y)$ is independent with $\pi$, we can solve Equation (3) based on the POMDP. We summarize the decision-making loop of the POMDP in Algorithm 1. Given a policy $\pi$, first, we initialize the $t = 0$, $T = p$ and $M_0 = \mathbf{0}$ (line 1). Next, we iteratively do sequential decision-making until the sampling process is completed (lines 2-10). In each iteration, the policy samples an action from the policy $\pi$ based on the current observation (line 3), and decides either to update the current state (line 6) or stop the trajectory (line 8) according to the action.

By framing the dynamic sampling challenge as a POMDP, we lay the groundwork for employing advanced Reinforcement Learning (RL) techniques, to derive an optimal policy for covariate selection. We solve Equation (3) based on our formulated POMDP via the Proximal Policy Optimization (PPO) algorithm (Schulman et al., 2017). We summarize the whole training process in Algorithm 2.

## 5 EXPERIMENTS

### 5.1 EXPERIMENTAL SETUP AND EVALUATION METRICS

**Datasets.** We explore our dynamic sampling strategy on two semi-synthetic datasets, *i.e.*, IHDP and ACIC, and a real-world dataset, *i.e.*, Jobs. The IHDP dataset (Hill, 2011) is constructed from the Infant Health and Development Program, which contains 747 samples and 25 covariates in total. The ACIC dataset is constructed from the Atlantic Causal Inference Conference competitions (Dorie et al., 2017), which includes 4,802 samples with 82 covariates. The JOBS dataset (LaLonde, 1986) is based on the National Supported Work program with 2,570 units (237 treated, 2,333 control) and 17 covariates from non-randomized observational studies. For all datasets, we randomly split the data into training set / testing set with ratios 9/1.

**CATE Estimation.** The goal of our dynamic sampling is to learn a policy that balances the cost of covariate selection and the accuracy of CATE estimation. In the training phase, we estimate $f_w(\mathscr{X}(X, \mathbf{1}))$ using widely used causal methods, namely TARNet (Shalit et al., 2017a), DESCN

---

**Algorithm 2** Counterfactual and Dynamic Policy Learning

---

**Require:** Dataset $\mathcal{D}$, weight parameter $\lambda$;

 1: Divide the observed dataset $\mathcal{D}$ into $\mathcal{D}_0$ and $\mathcal{D}_1$;
 2: **for** $i \leftarrow 1$ to max iteration step **do**
 3:    Sample $\mathcal{D}_0^{\text{batch}}$ and $\mathcal{D}_1^{\text{batch}}$ from $\mathcal{D}_0$ and $\mathcal{D}_1$;
 4:    Compute the gradients w.r.t $f_0$ and $f_1$;
 5:    Upgrade $f_0$ and $f_1$ via stochastic gradient descent;
 6: **end for**
 7: **for** $(X, W, Y) \in \mathcal{D}$ **do**
 8:    Compute $\hat{Y}(1 - w) = f_w(\mathscr{X}(X, \mathbf{1}))$;
 9: **end for**
10: Initialize policy $\pi_\theta$ and old policy $\pi_{\theta_{\text{old}}} \leftarrow \pi$;
11: **for** $i \leftarrow 1$ to max iteration step **do**
12:    Run $\pi_{\theta_{\text{old}}}$ (Algorithm 1) to sample a series state and action pairs $(\mathscr{X}(X, M_t), a)$ from trajectories;
13:    Update $\theta$ via PPO's loss;
14: **end for**

**Output:** estimation models $f_0$ and $f_1$ and policy $\pi$.

---

Table 2: Performance comparison of the cost, causal effect ($\sqrt{\epsilon_{\text{PEHE}}}$ or $R_{\text{Pol}}$) and Total under $\lambda = 1$ on three dataset IHDP, ACIC and Jobs. The best results are bolded.

| | IHDP ($\lambda = 1$) | | | ACIC ($\lambda = 1$) | | | Jobs ($\lambda = 100$) | | |
|---|---|---|---|---|---|---|---|---|---|
| | Cost ($\downarrow$) | $\sqrt{\epsilon_{\text{PEHE}}}$ ($\downarrow$) | Total ($\downarrow$) | Cost ($\downarrow$) | $\sqrt{\epsilon_{\text{PEHE}}}$ ($\downarrow$) | Total ($\downarrow$) | Cost ($\downarrow$) | $R_{\text{Pol}}$($\downarrow$) | Total ($\downarrow$) |
| Random | 15.51±9.27 | 7.58±0.51 | 73.23±7.74 | 52.36±23.18 | 9.90±1.68 | 153.20±32.04 | 12.13±5.40 | 0.15±0.01 | 27.44±1.15 |
| Static (TARNet) | 3.12±1.69 | 7.81±0.45 | 64.32±6.86 | 42.86±1.64 | 5.57±1.23 | 75.40±14.45 | 4.00±0.00 | 0.21±0.06 | 25.59±5.51 |
| Greedy (TARNet) | 5.37±1.58 | 7.73±0.70 | 65.57±11.15 | 1.88±0.63 | 5.82±2.32 | 41.07±32.61 | 1.44±0.60 | 0.18±0.05 | 19.68±4.26 |
| Ours (TARNet) | 9.96±1.30 | 6.90±0.60 | **57.97±7.71** | 0.96±0.41 | 4.94±1.69 | **28.26±19.11** | 0.85±0.85 | 0.18±0.08 | **18.98±5.77** |
| Oracle (TARNet) | 9.64±0.91 | 5.96±0.55 | 45.43±6.21 | 1.47±0.25 | 5.11±1.39 | 29.54±11.25 | 1.51±0.37 | 0.11±0.07 | 13.23±5.26 |
| Static (DESCN) | 2.75±1.09 | 7.83±0.52 | 64.29±7.73 | 42.57±1.92 | 5.32±0.94 | 71.81±10.65 | 0.12±0.33 | 0.22±0.05 | 22.90±4.67 |
| Greedy (DESCN) | 5.03±2.46 | 7.84±0.77 | 67.07±10.67 | 1.93±0.60 | 5.19±2.22 | 33.77±31.76 | 1.05±0.13 | 0.18±0.05 | 18.57±4.30 |
| Ours (DESCN) | 9.64±1.83 | 6.86±0.54 | **56.97±6.88** | 0.65±0.85 | 4.85±1.54 | **26.58±17.73** | 0.64±0.50 | 0.17±0.07 | **17.77±5.65** |
| Oracle (DESCN) | 9.15±0.88 | 6.00±0.54 | 45.50±6.27 | 1.50±0.26 | 5.05±1.39 | 28.99±11.02 | 1.00±0.01 | 0.11±0.02 | 12.05±1.31 |
| Static (ESCFR) | 3.12±1.05 | 7.82±0.48 | 64.56±7.59 | 41.86±2.17 | 5.13±0.85 | 68.89±9.45 | 3.75±1.56 | 0.18±0.06 | 21.47±4.00 |
| Greedy (ESCFR) | 4.03±1.65 | 7.87±0.68 | 66.44±10.25 | 1.89±0.49 | 5.45±2.42 | 37.47±34.69 | 1.71±0.46 | 0.14±0.05 | 15.80±3.56 |
| Ours (ESCFR) | 9.38±1.32 | 6.78±0.61 | **56.60±6.72** | 0.94±0.41 | 4.49±0.90 | **21.93±8.58** | 1.39±0.82 | 0.13±0.08 | **15.13±5.86** |
| Oracle (ESCFR) | 9.73±0.99 | 5.94±0.57 | 45.35±6.28 | 1.37±0.19 | 4.93±1.20 | 27.10±9.97 | 1.69±0.51 | 0.08±0.03 | 9.68±1.58 |
| Static (C. Forest) | 2.25±1.39 | 7.91±0.57 | 65.19±8.20 | 41.86±0.64 | 5.07±0.66 | 68.00±6.61 | 4.00±0.00 | 0.23±0.08 | 27.31±8.43 |
| Greedy (C. Forest) | 3.30±1.89 | 7.97±0.70 | 67.37±10.86 | 1.80±0.49 | 4.88±1.11 | 26.87±10.96 | 1.37±0.43 | 0.20±0.06 | 21.82±6.07 |
| Ours (C. Forest) | 9.79±1.66 | 6.99±0.55 | **58.93±7.20** | 1.30±0.89 | 4.35±0.96 | **21.12±8.03** | 0.49±0.60 | 0.20±0.07 | **20.74±6.46** |
| Oracle (C. Forest) | 9.32±0.88 | 5.94±0.58 | 44.93±6.80 | 1.56±0.37 | 2.37±0.39 | 7.32±2.15 | 1.57±0.47 | 0.14±0.02 | 15.17±1.32 |
| Static (DeRCFR) | 2.00±1.94 | 8.01±0.55 | 66.5±7.25 | 42.14±0.64 | 5.29±0.85 | 70.87±9.23 | 3.75±0.66 | 0.23±0.06 | 26.92±6.24 |
| Greedy (DeRCFR) | 4.40±1.69 | 7.80±0.64 | 65.70±9.19 | 1.98±0.38 | 4.63±0.91 | 24.23±9.66 | 1.28±0.42 | 0.18±0.05 | 19.73±4.92 |
| Ours (DeRCFR) | 10.25±1.72 | 6.94±0.58 | **58.72±7.35** | 1.69±0.60 | 4.2±0.39 | **19.45±3.77** | 0.82±0.69 | 0.18±0.07 | **19.29±6.14** |
| Oracle (DeRCFR) | 9.88±0.89 | 5.88±0.60 | 44.85±6.87 | 1.58±0.37 | 2.53±0.56 | 8.31±3.38 | 1.43±0.45 | 0.15±0.07 | 16.65±6.89 |
| Static (DN) | 1.88±1.27 | 7.95±0.65 | 65.44±8.91 | 42.29±1.03 | 5.46±0.96 | 73.00±10.85 | 3.75±0.66 | 0.21±0.06 | 24.65±6.43 |
| Greedy (DN) | 4.06±2.97 | 7.89±0.71 | 66.78±9.24 | 1.80±0.45 | 4.67±1.01 | 24.59±10.89 | 1.39±0.44 | 0.17±0.04 | 17.96±3.89 |
| Ours (DN) | 9.81±1.93 | 7.01±0.60 | **59.26±7.71** | 1.09±0.78 | 4.39±0.69 | **21.31±8.56** | 0.87±1.07 | 0.17±0.07 | **17.62±6.01** |
| Oracle (DN) | 9.40±1.10 | 5.94±0.56 | 45.00±6.68 | 1.52±0.35 | 2.01±0.28 | 5.65±1.47 | 1.61±0.55 | 0.15±0.06 | 16.16±5.52 |

(Zhong et al., 2022), ESCFR (Wang et al., 2023), Causal Forest (Wager & Athey, 2018), DeRCFR (Wu et al., 2020), and DragonNet (Shi et al., 2019).

**Metrics.** Our evaluation also consists of accumulative costs and the accuracy of causal effects. For cost, we calculate the sum of all costs of measured index $C = \sum_{a \in \mathcal{A}(M)} c(a)$ for each test data. For causal effects, we calculate $\sqrt{\epsilon_{\text{PEHE}}} = \sqrt{\frac{1}{|\mathcal{D}|} \sum_{(X, Y(0), Y(1)) \sim \mathcal{D}} ((f_1 - f_0) - (Y(1) - Y(0)))^2}$ for the IHDP and the ACIC dataset (which can access to the ground truth potential outcomes) to measure the accuracy of the estimated CATE based on the partially observed covariates, where $f_w = f_w(\mathscr{X}(X_i, M))$ for $w = 0, 1$. For the Jobs (which can not access to the $Y(1 - W)$), we calculate $R_{\text{Pol}} = 1 - (\mathbb{E}[Y(1) \mid f_1 - f_0 > 0, T = 1] \cdot \mathbb{P}(f_1 - f_0 > 0) + \mathbb{E}[Y(0) \mid \hat{f}_1 - f_0 \leq 0, T = 0] \cdot \mathbb{P}(f_1 - f_0 \leq 0))$, where $T$ is the treatment indicator. To comprehensively evaluate a sampling policy, we sum the cost and the inaccuracy of the CATE estimation as in Equation (3), called *Total*, representing the cost-accuracy trade-off.

**Cost function.** We consider two types of cost functions. The one is *all-one cost* that $c \equiv 1$, a basic setting where the costs of each covariate are the same. The other is *relative cost*, which is more practical where the covariate more correlated with the outcome will have a higher observed cost. In

Table 3: Performance comparison on the IHDP, ACIC, and Jobs datasets. The cost function is positively correlated with the correlation coefficients of covariates and outcome ($\alpha = 1$).

| | IHDP ($\lambda = 1$) | | | ACIC ($\lambda = 1$) | | | Jobs ($\lambda = 100$) | | |
|---|---|---|---|---|---|---|---|---|---|
| | Cost ($\downarrow$) | $\sqrt{\epsilon_{\text{PEHE}}}$ ($\downarrow$) | Total ($\downarrow$) | Cost ($\downarrow$) | $\sqrt{\epsilon_{\text{PEHE}}}$ ($\downarrow$) | Total ($\downarrow$) | Cost ($\downarrow$) | $R_{\text{Pol}}$($\downarrow$) | Total ($\downarrow$) |
| Random | 2.37±1.42 | 7.59±0.49 | 60.27±7.47 | 3.18±1.80 | 9.90±1.68 | 104.03±32.05 | 2.48±1.39 | 0.15±0.01 | 17.80±0.76 |
| Static (TARNet) | 2.22±0.46 | 7.39±0.62 | 57.16±9.08 | 2.56±0.65 | 5.66±0.92 | 35.39±10.27 | 0.79±0.24 | 0.18±0.05 | 19.25±4.42 |
| Greedy (TARNet) | 1.26±0.32 | 7.70±0.72 | 61.05±11.74 | 0.37±0.35 | 5.15±1.69 | 29.76±20.41 | 0.16±0.08 | 0.17±0.05 | 17.20±4.16 |
| Ours (TARNet) | 2.63±0.60 | 6.84±0.63 | **49.67±8.47** | 0.14±0.10 | 4.64±0.81 | **22.38±7.61** | 0.11±0.10 | 0.17±0.07 | **16.24±5.23** |
| Oracle (TARNet) | 2.17±0.38 | 5.72±0.59 | 35.24±6.47 | 0.31±0.22 | 5.08±1.39 | 28.07±10.92 | 0.24±0.09 | 0.10±0.05 | 9.97±2.90 |
| Static (DESCN) | 2.10±0.52 | 7.42±0.58 | 57.46±8.64 | 2.64±0.50 | 5.72±0.77 | 35.90±8.71 | 0.31±0.17 | 0.18±0.07 | 19.21±6.01 |
| Greedy (DESCN) | 1.21±0.65 | 7.67±0.73 | 60.65±11.43 | 0.39±0.36 | 4.73±1.37 | 24.38±16.61 | 0.15±0.06 | 0.15±0.04 | 15.36±3.06 |
| Ours (DESCN) | 2.66±0.58 | 6.83±0.62 | **49.62±8.49** | 0.15±0.08 | 4.60±0.79 | **21.89±7.28** | 0.08±0.07 | 0.15±0.03 | **15.28±2.05** |
| Oracle (DESCN) | 2.14±0.39 | 5.72±0.60 | 35.17±6.56 | 0.29±0.22 | 4.87±1.20 | 25.47±9.70 | 0.13±0.04 | 0.09±0.02 | 9.34±0.95 |
| Static (ESCFR) | 1.89±0.60 | 7.36±0.53 | 56.33±7.78 | 2.55±0.74 | 5.65±1.02 | 35.84±11.66 | 0.87±0.28 | 0.16±0.04 | 17.17±3.06 |
| Greedy (ESCFR) | 0.97±0.42 | 7.85±0.72 | 63.13±11.78 | 0.35±0.31 | 4.65±1.30 | 23.63±14.82 | 0.25±0.06 | 0.14±0.04 | 13.98±2.96 |
| Ours (ESCFR) | 2.47±0.60 | 6.80±0.58 | **49.50±8.39** | 0.21±0.17 | 4.57±0.72 | **21.57±6.71** | 0.21±0.14 | 0.13±0.07 | **13.94±5.47** |
| Oracle (ESCFR) | 2.18±0.38 | 5.71±0.60 | 35.18±6.54 | 0.32±0.23 | 5.02±1.38 | 27.42±10.65 | 0.28±0.11 | 0.08±0.03 | 7.92±1.44 |
| Static (C. Forest) | 1.98±0.55 | 7.52±0.59 | 58.81±8.95 | 2.65±0.63 | 5.14±0.76 | 29.60±7.71 | 0.56±0.23 | 0.22±0.06 | 22.26±6.10 |
| Greedy (C. Forest) | 0.76±0.52 | 7.94±0.72 | 64.35±11.62 | 0.29±0.28 | 4.44±1.15 | 21.36±11.26 | 0.21±0.12 | 0.20±0.05 | 20.05±5.22 |
| Ours (C. Forest) | 2.15±0.60 | 6.93±0.61 | **50.55±8.44** | 0.42±0.43 | 4.38±1.02 | **20.67±10.19** | 0.25±0.14 | 0.16±0.06 | **16.41±6.14** |
| Oracle (C. Forest) | 2.11±0.39 | 5.68±0.64 | 34.81±7.00 | 0.30±0.27 | 1.92±0.20 | 4.04±1.00 | 0.25±0.09 | 0.12±0.03 | 12.57±2.87 |
| Static (DeRCFR) | 1.97±0.56 | 7.52±0.57 | 58.85±8.65 | 2.58±0.52 | 5.38±0.78 | 32.19±8.14 | 0.69±0.26 | 0.24±0.07 | 24.79±6.87 |
| Greedy (DeRCFR) | 1.05±0.38 | 7.75±0.68 | 61.58±10.64 | 0.46±0.43 | 4.88±1.60 | 26.88±17.29 | 0.18±0.11 | 0.17±0.05 | 17.36±4.85 |
| Ours (DeRCFR) | 2.27±0.44 | 6.78±0.49 | **48.43±6.60** | 0.48±0.43 | 4.19±0.73 | **18.43±5.26** | 0.18±0.08 | 0.17±0.07 | **17.30±6.61** |
| Oracle (DeRCFR) | 2.17±0.38 | 5.67±0.65 | 34.78±7.04 | 0.31±0.28 | 2.46±0.59 | 6.69±3.46 | 0.26±0.13 | 0.14±0.07 | 13.97±6.90 |
| Static (DN) | 2.00±0.54 | 7.51±0.59 | 58.75±8.90 | 2.65±0.44 | 5.62±0.90 | 35.03±9.48 | 0.64±0.27 | 0.22±0.05 | 22.25±5.20 |
| Greedy (DN) | 0.84±0.57 | 7.85±0.69 | 62.98±10.45 | 0.43±0.42 | 4.87±1.54 | 26.53±16.79 | 0.21±0.10 | 0.16±0.05 | 16.65±4.74 |
| Ours (DN) | 2.35±0.49 | 6.86±0.52 | **49.67±7.07** | 0.48±0.31 | 4.09±0.53 | **17.49±3.96** | 0.3±0.31 | 0.15±0.05 | **16.53±4.36** |
| Oracle (DN) | 2.11±0.40 | 5.69±0.63 | 34.86±6.92 | 0.30±0.27 | 2.29±0.39 | 5.69±2.03 | | | |

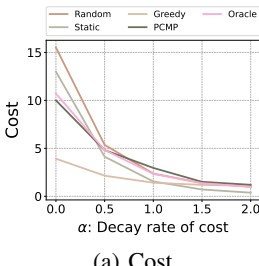
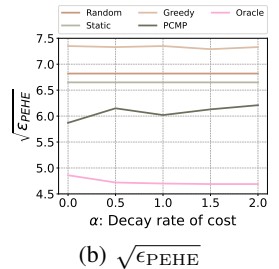
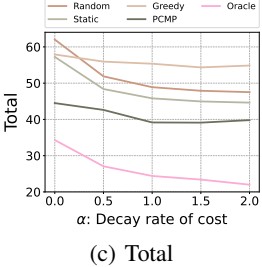

(a) Cost     (b) $\sqrt{\epsilon_{\text{PEHE}}}$     (c) Total

Figure 1: Performance of four baseline methods and our PCMP on the IHDP with varying $\alpha$.

particular, we calculate the absolute value of the correlation coefficients of each covariate and the outcome and sort them. According to the correlation coefficients from large to small, we assign the cost to $1, 1/2^{\alpha}, \cdots, 1/p^{\alpha}$. Note when $\alpha = 0$, it degenerates into all-one cost.

## 5.2 BASELINE SAMPLING POLICES.

We compare our dynamic sampling policy with the following four baseline sampling policies.

**Random**, a non-data-driven sampling policy that randomly chooses to stop or continue sampling, and if to continue, randomly measures a covariate in each acquisition step.

**Static**, a data-driven sampling policy that uses a fixed mask vector $M^*$ to act on each data without randomness. In practice, we get $M^*$ by solving Equation (4) via grid search.

**Greedy**, a data-driven and adaptive sampling policy. In practice, we find out the $k$-nearest neighborhood of the measured covariates in the training dataset and then select the unmeasured covariate that leads to the greatest average Total decreasing within these $k$ neighbors. We stop when no unmeasured covariate will lead to the average Total decrease.

**Oracle Greedy**, a theoretical benchmark sampling policy that provides a lower bound on the Total. In practice, We select the unmeasured covariate that leads to the greatest Total decrease in each acquisition step for each test data. We stop when no unmeasured covariate will lead to the Total decrease. It's important to reiterate that this policy is not a feasible method in practical scenarios due to its reliance on foresight, and also unfair to compare it with other policies.

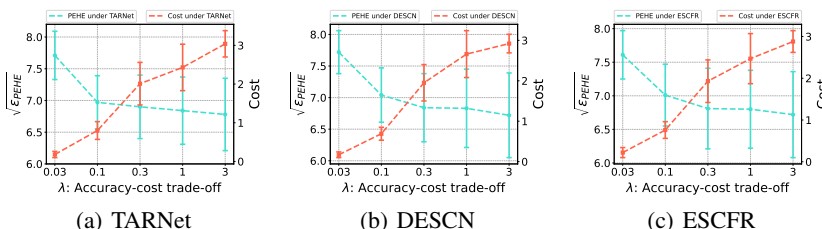

Figure 2: Performance of cost and $\sqrt{\epsilon_{\mathrm{PEHE}}}$ with varying weight $\lambda$ on the IHDP under $\alpha = 1$.

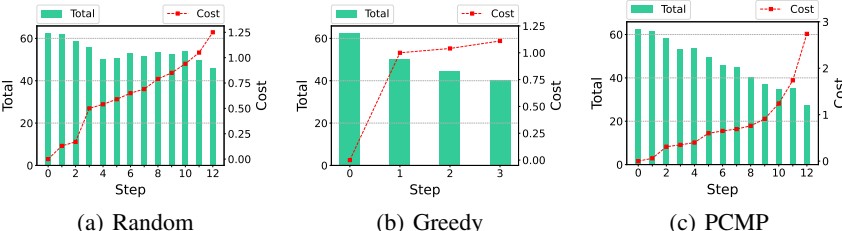

Figure 3: Visualizations of test data from the IHDP dataset under $\alpha = 1$ across the acquisition steps.

### 5.3 PERFORMANCE COMPARISON

We compare our dynamic sampling policy and baseline method in the IHDP, ACIC, and Jobs datasets under all-one cost setting. The results are shown in Table 2. The PCMP obtains a lower Total than other baselines and even outperforms oracle's method in some cases.

**Effects of Cost Function.** We further explore the effects of cost function $c$. We show the results of $\alpha = 1$ in Table 3. Our proposed PCMP outperforms other fair baseline methods in the case where the measurement cost is positively correlated with the correlation between covariates and the outcome. Furthermore, we show the cost, $\sqrt{\epsilon_{\mathrm{PEHE}}}$ and the Total over $\alpha = 0, 0.5, 1, 1.5, 2$ in Figure 1. These results illustrate that the proposed PCMP can achieve optimal performance by adaptively choosing the measurement order and combination according to the different cost functions.

**Effects of Weight Parameter.** We further study the effects of weight parameter $\lambda$ on our dynamic policy. We show the accumulative cost and $\sqrt{\epsilon_{\mathrm{PEHE}}}$ over $\lambda = 0.03, 0.1, 0.3, 1, 3$ on IHDP under $\alpha = 1$, and the results are shown in Figure 2. It shows that as lambda increases, the cost gradually increases while $\sqrt{\epsilon_{\mathrm{PEHE}}}$ gradually decreases. The results are consistent with our intuitive understanding of the optimisation objective Total and provide guidance for reality. In practice, we can tune the weight parameter $\lambda$ to the preference of our policy–lower cost or higher accuracy.

### 5.4 IN-DEPTH ANALYSIS

We visualize the cost, PEHE, and Total per acquisition step for the random, greedy, and our dynamic sampling policy on test data from the IHDP, and the results are shown in Figure 3. The Total of the random policy decreases slowly with slight oscillations as the acquisition step increases, while the Total of the greedy policy decreases the most rapidly but stops sampling early. Unlike the two baseline methods above, the Total of our proposed PCMP declines smoothly and quite rapidly and allows for a longer sampling process, eventually reaching the lowest Total. This demonstrates that our dynamic sampling policy is a fore-sighted policy.

## 6 CONCLUSION

In this work, we discuss a novel treatment effect estimation problem, *i.e.*, how to balance measurement cost and accuracy in the case of incomplete observation of all covariates. We extend previous methodologies for estimating treatment effects, introducing the capability to handle scenarios where covariates are partially observed. Then, we introduce the dynamic covariate measurement policy which adaptively decides which covariate to measure or stop sampling at each acquisition step according to the observed covariates. We further show that our dynamic sampling policy is superior to other baseline policies theoretically and empirically.

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

IMPACT STATEMENT

This paper introduces the Active Causal Learning (ACL) framework for heterogeneous treatment effect estimation, a significant step forward in the application of machine learning for causal inference. The technique is particularly beneficial in the healthcare sector, where it can aid in discerning the causal determinants of diseases from sub-sampled series data, facilitating the creation of more effective treatment plans. The PCMP allows for a dynamic and systematic approach to modeling, which can lead to better-informed medical decisions and personalized patient care. However, its effectiveness is contingent upon the quality and structure of the data; if the underlying data fails to capture the true complexity of causal relationships or exhibits inconsistencies, the model's capacity to accurately identify these relationships may be compromised. Despite these limitations, the potential impact of this research in improving the balance of measurement cost and health outcomes through more nuanced data analysis is substantial.

## A  MORE EXPERIMENTAL DETAILS

### A.1  OUR METHODS

The *Estimation Model* is designed for causal inference, processing input data through a customizable data preprocessing layer followed by a Multi-Layer Perceptron (MLP) architecture. The data preprocessing layer, which can be selected from multiple versions, handles both numerical and categorical data, incorporating embeddings for categorical variables and accounting for missing data. The processed data is then fed into two separate MLPs based on the treatment variable. Each MLP, consisting of an input layer, a hidden layer with LeakyReLU activation and dropout, and an output layer, predicts potential outcomes $y(0)$ or $y(1)$ for the respective subsets of the data. This architecture allows for efficient and specialized processing of data for causal inference applications.

We train the estimation model by minimizing the MSE loss via Adam optimizer. The learning rate is set to 0.001, the weight decay is set to 0.00001, and the embedding size is set to 2. We train is for 50 epochs. The choice of mask vector is quite tricky. We provide the mask vector sampling strategy called *uniform*, which indicates that we first uniformly sample the number of 1 in the mask vector, and then uniformly sample a mask vector $M$ that $M^{\top} \cdot \mathbf{1}$ equals to the number.

The *Policy Model* in this setup utilizes the data preprocessing layer from the Estimation Model for feature extraction. This extractor processes input observations and passes them through a fully connected neural network, which consists of three layers with LeakyReLU activations behind the beginning two linear layers, to generate features for policy decisions. Additionally, the Policy Model includes an Action Network that modifies these features based on the presence of missing data, indicated by a mask. This network emphasizes relevant features and diminishes the impact of missing data, ultimately producing an output that informs policy decisions in a context-sensitive manner.

Inspired by the PPO algorithm, we solve the dynamic sampling optimization problem based on our formulated POMDP via policy gradient. Specifically, given a parameterized policy $\pi_\theta$, its value function is defined as

$$V_{\pi_\theta}(\mathscr{X}(X, M_t)) = \mathbb{E}_\tau \left[ \sum_{u=t}^{T} G(\tau) \mid s_t = \mathscr{X}(X, M_u), \tau \sim \pi^\theta \right],$$

which is the expected sum of rewards of all the trajectories when the agent starts at an intermediate state $\mathscr{X}(X, M_t)$ and then follows $\pi$; its Q function is defined as

$$Q_{\pi_\theta}(\mathscr{X}(X, M_t), a) = \mathbb{E}_\tau \left[ \sum_{u=t}^{T} G(\tau) \mid s_t = \mathscr{X}(X, M_u), a_{t+1} = a, \tau \sim \pi^\theta \right].$$

which is the expected sum of rewards of all the trajectories when the agent starts at an intermediate state $\mathscr{X}(X, M_t)$, takes action $a$, and then follows $\pi$. Then, we define its advantage function as

$$A_{\pi_\theta}(\mathscr{X}(X, M_t), a) = Q_{\pi_\theta}(\mathscr{X}(X, M_t), a) - V_{\pi_\theta}(\mathscr{X}(X, M_t)).$$

to quantify how good it is if we take action $a$ other than other actions at $\mathscr{X}(X, M_t)$. Furthermore, to penalise the selected covariates that do not perform well while rewarding the one that gets a high sum

of rewards, we keep updating an old policy $\pi_{\theta_{\mathrm{old}}}$ and define the baseline reward as

$$b_\theta(\mathscr{X}(X, M_t), a) = \frac{\pi_\theta(a \mid \mathscr{X}(X, M_t))}{\pi_{\theta_{\mathrm{old}}}(a \mid \mathscr{X}(X, M_t))}.$$

We follow policy gradient methods (Sutton et al., 1999; Silver et al., 2014) to optimize $\pi_\theta$ with respect to the following surrogate objective:

$$J(\theta) = \mathbb{E}_{(\mathscr{X}(X, M_t), a) \sim \pi_{\theta_{\mathrm{old}}}} \left[ \min\{b_\theta \cdot A_{\pi_{\theta_{\mathrm{old}}}}, \mathrm{clip}(b_\theta, 1 - \epsilon, 1 + \epsilon)\} \cdot A_{\pi_{\theta_{\mathrm{old}}}} \right] \quad (5)$$

where $\mathrm{clip}(x, a, b) = \max\{\min\{x, b\}, a\}$, $\epsilon$ is a hyperparameter controlling the clipping extent, and the input of $b_\theta$ and $A_{\pi_{\theta_{\mathrm{old}}}}$ is $(\mathscr{X}(X, M_t), a)$. This training process iteratively refines the policy $\pi_\theta$, while the old policy $\pi_{\theta_{\mathrm{old}}}$ is synchronized with the current policy $\pi_\theta$ after a specified number of iterations.

We implement the *PPO algorithm* based on the stable-baselines3 (Raffin et al., 2021) which is a popular framework for reliable implementations of RL algorithms. The learning rate is set to 0.001, the discount factor $\gamma$ is set to 0.99, and the other hyperparameters are the default values. We train our policy for 50,000 steps. The initialization of the policy is quite tricky. When the weight parameter $\lambda$ is quite small, which indicates that the optimal sampling trajectory will be short, we add a learnable parameter to increase the probability of $0 \sim \pi_\theta$.

We refer you to our official code for more details.

### A.2 BASELINE METHODS

### A.2.1 RANDOM POLICY

The Random Policy is implemented as a baseline strategy for variable selection in a non-deterministic and non-adaptive manner. At each acquisition step, the policy randomly decides whether to stop sampling or continue. The decision to stop or continue is made using a random generator with a uniform distribution. If the decision is to continue, it randomly selects an unmeasured covariate from the available set. The process iterates until a predefined stopping criterion.

We summarize the process of random policy in Algorithm 3.

---

**Algorithm 3** Random Policy for Variable Selection

---

**Require:** test data $X$ and a threshold $\mu$
1: Initialize the mask vector $M = \mathbf{0}$;
2: **while** True **do**
3:    Decide randomly to stop or continue $p \sim \mathrm{Uniform}[0, 1]$;
4:    **if** $p < \mu$ **then**
5:       **break**
6:    **else**
7:       Randomly sample $a \sim \mathrm{Uniform}(\{1, 2, \cdots, n\} \setminus \mathcal{A}(M))$;
8:       Update the mask vector $M \leftarrow M + e_a$;
9:    **end if**
10: **end while**
11: Output the measured covariates $\mathscr{X}(X, M)$.

---

We choose $\mu = 1/p$.

### A.2.2 OPTIMAL STATIC

The Static Policy is also a baseline strategy for variable selection in a deterministic, data-driven, and non-adaptive manner. It is a two-stage approach designed for variable selection, focusing on the generation and application of an optimal mask vector. This method diverges from dynamic selection strategies by employing a static, uniform approach to variable sampling across all test instances.

The first stage involves training on a given dataset to derive an optimal mask vector by solving Equation (4). This vector represents a fixed pattern of variable selection, determined based on the dataset's characteristics and the target objective.

Since the objective function of Equation (4) is non-differentiable to $M$, we leverage the grid search algorithm, a derivative-free optimization method to solve the optimization problem. This is achieved by iteratively testing different combinations of variables, represented by a mask vector, and evaluating their performance based on a Total computed by the environment. The Total is calculated for each possible mask vector by turning one of the dimensions on or off in the mask and observing the effect on the model's output. The search for the optimal mask involves computing Totals for all variables and updating the current mask based on which variable leads to the minimum Total. If the addition of a new mask vector does not improve the minimum Total by a significant margin, the search terminates, and the current mask is considered optimal.

The second stage is the inference stage. Once the optimal mask vector is obtained, it is applied uniformly across all test data instances. This means that every test instance is evaluated using the same set of variables, as dictated by the mask vector.

We summarize the two-stage process in Algorithm 4.

---

**Algorithm 4** Static Policy for Variable Selection

---

**Require:** Training dataset $\mathcal{D}$
1: Randomly initialize the mask vector $M$, initialize $S_{\text{opt}} \leftarrow \infty$;
2: **for** $step = 1$ to Max Number of Step **do**
3:     Initialize an empty list $Totals$;
4:     **for** $a = 1$ to $p$ **do**
5:         $M(a) \leftarrow 1 - M(a)$;
6:         Sample a mini-batch from $\mathcal{D}$, denoted as $\mathcal{D}^{\text{batch}}$;
7:         Calculate the average Total;
8:         Append the average Total to $Totals$;
9:         $M(a) \leftarrow 1 - M(a)$;
10:     **end for**
11:     Find index $a$ of the minimum average Total in $Totals$;
12:     **if** $S_{\text{opt}} > Totals[a]$ **then**
13:         Update the current mask $M(a) \leftarrow 1 - M(a)$;
14:         Update the optimal Total $S_{\text{opt}} \leftarrow Totals[a]$;
15:     **else**
16:         **break**
17:     **end if**
18: **end for**
19: Output the measured covariates $\mathscr{X}(X, M)$.

---

### A.2.3 GREEDY POLICY

The Greedy Policy is designed as a heuristic approach for variable selection in a data-driven manner. It selects variables by integrating the k-nearest neighbors (KNN) approach with a greedy selection mechanism.

Initially, for each test instance, the algorithm identifies its k-nearest neighbors within the training dataset. In particular, for quantitative variables, we first standardize the value of the dimension in the data set to [0, 1], and then define the distance as the absolute value of the difference; for categorical variables, we define the distance of different categories as 1 and the distance of the same category as 0. This identification is based on the Euclidean distance metric. The number of neighbors, denoted as $k$, is a critical parameter and its optimal value is determined through experimental tuning.

Once the nearest neighbors are identified, the algorithm enters an iterative covariate evaluation phase. In each iteration, it assesses the impact of each unmeasured covariate on the predictive model's performance. This is achieved by temporarily including each covariate in the model and calculating the resulting average Total change across the k-nearest neighbors.

The core of the Greedy Policy lies in its selection criterion. In every iteration, the algorithm selects the unmeasured covariate that yields the highest average Total decrease. This covariate is then permanently added to the set of selected variables, and the sampling pattern is updated accordingly.

This process of evaluating and adding covariates continues iteratively. After each iteration, the algorithm re-evaluates the remaining unmeasured covariates, as the inclusion of a new covariate can change the dynamics of the model's performance. The algorithm halts when there are no more covariates that significantly improve the model's Total, indicating that the addition of further variables would likely not provide substantial benefits.

We summarize the whole process of greedy policy for specific test data in Appendix A.2.3.

---

**Algorithm 5** Greedy Policy for Variable Selection

---

**Require:** Training data set $\mathcal{D}$, a test data $(X, W, Y)$, a scoring function $S$, number of neighbors $k$, and an improvement threshold $\beta$
1: Initialize the mask vector $M \leftarrow \mathbf{0}$ and the optimal Total $S_{\text{opt}} \leftarrow \infty$;
2: **while** True **do**
3:     Initialize the selected variable $a_{\text{selected}} \leftarrow$ null;
4:     Determine the k-nearest neighbors of the test data in $\mathcal{D}$, denoted as $\mathcal{D}^{\text{neighbor}}$;
5:     **for** each unmeasured index $a$ in $\{1, 2, \cdots, n\} \setminus \mathcal{A}(M)$ **do**
6:         Update the mask vector $M_a \leftarrow M + e_a$;
7:         Calculate the average Total improvement of $\mathcal{D}^{\text{neighbor}}$;
8:         **if** $S_{\text{opt} \cdot \beta > S_a}$ **then**
9:             Update $S_{\text{opt}} \leftarrow S_a$;
10:            Update $a_{\text{selected}} \leftarrow a$;
11:         **end if**
12:     **end for**
13:     **if** $a_{\text{selected}}$ is null **then**
14:         **break**
15:     **end if**
16:     Update the mask vector $M \leftarrow M + e_{a_{\text{selected}}}$;
17: **end while**
18: Output the measured covariates $\mathscr{X}(X, M)$.

---

We choose $\beta = 1$ and $k = 5$.

## A.2.4 ORACLE GREEDY

The Oracle Greedy Policy is an idealized, theoretical approach to variable selection that assumes access to perfect, omniscient knowledge about the impact of each covariate on the model's performance. Unlike other methods, which relies on data-driven estimations and heuristics, the Oracle Greedy uses its 'all-knowing' perspective to make the most optimal choices at each step.

At each iteration of the variable selection process, the Oracle reviews all the unmeasured covariates. With its perfect foresight, the Oracle predicts the exact change in the score – as defined by the objective function of Equation (3) – that would result from the inclusion of each covariate. Similar to the selection criterion of Greedy Policy, it then selects the covariate that offers the most significant decrement in the score. What's more, the Oracle reviews can also achieve the ground truth value of $y(0)$ and $y(1)$, so the score is calculated by the ground truth values, instead of the estimated values via some causal methods.

The process continues iteratively, with the Oracle selecting the most impactful covariate at each step. The termination of the algorithm occurs when adding any of the remaining covariates ceases to significantly improve the model's score. However, this stopping criterion, like the selection process itself, is based on the Oracle's perfect knowledge rather than on empirical data analysis or a significant improvement threshold.

**Notation.** It's important to reiterate that the Oracle Greedy Policy is not a feasible method in practical scenarios due to its reliance on an unrealistic level of foresight. It serves as a theoretical benchmark, providing an upper bound on the efficacy of variable selection strategies. This conceptual tool allows researchers to gauge the potential limits of their data-driven methods and to understand the gap between practical algorithms and the idealized 'perfect' selection strategy.

We summarize the oracle process for specific test data in Appendix A.2.4 with $\beta = 1$.

**Algorithm 6** Oracle Greedy Policy for Variable Selection

---

**Require:** Test data $(X, Y(0), Y(1))$, two estimation models $f_0$ and $f_1$, an oracle scoring function $S$
1: Initialize the mask vector $M = \mathbf{0}$ and the optimal score $S_{\text{opt}} \leftarrow -\infty$
2: **while** True **do**
3: Initialize the selected variable $a_{\text{selected}} \leftarrow$ null
4: **for** each unmeasured index $a$ in $\{1, 2, \cdots, n\} \setminus \mathcal{A}(M)$ **do**
5:  Update the mask vector $M_a = M + e_a$
6:  Calculate the oracle score $S_a \leftarrow S(x, y(0), y(1), M)$
7:  **if** $S_a \cdot \beta > S_{\text{opt}}$ **then**
8:   $S_{\text{opt}} \leftarrow S_a$
9:   $a_{\text{selected}} \leftarrow a$
10:  **end if**
11: **end for**
12: **if** $a_{\text{selected}}$ is null **then**
13:  **break**
14: **end if**
15: Update the mask vector $M \leftarrow M + e_{a_{\text{selected}}}$
16: **end while**
17: Output the measured covariates $\mathscr{X}(X, M)$.

---

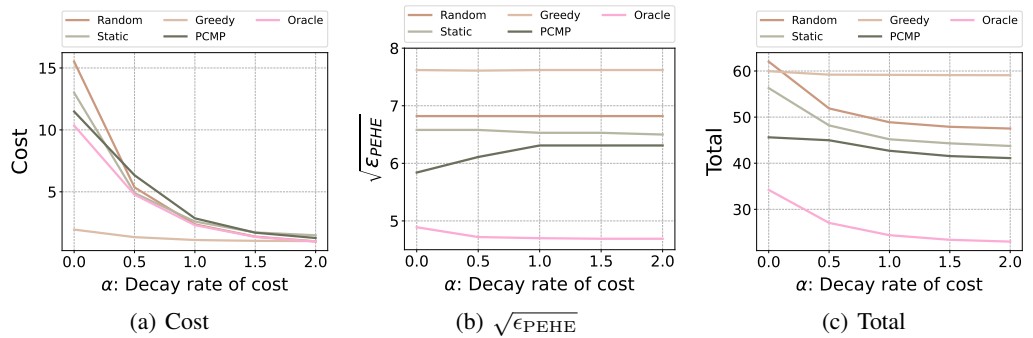

(a) Cost     (b) $\sqrt{\epsilon_{\text{PEHE}}}$     (c) Total

Figure 4: Comparative analysis of four baseline methods and the proposed PCMP on the IHDP. We leverage the causal method TARNet. We show the cost, $\sqrt{\epsilon_{\text{PEHE}}}$ and Total across $\alpha = 0, 0.5, 1.0, 1.5, 2.0$.

# B  FURTHER EXPERIMENTS

## B.1  ADDITIONAL EXPERIMENT RESULTS

**Effects of Cost Function.** We show the cost, $\sqrt{\epsilon_{\text{PEHE}}}$ and Total across $\alpha = 0, 0.5, 1, 1.5, 2$ in Figure 4 (for TARNet) and Figure 5 (for DESCN). Similar results can be seen. This results reinforce the fact that our PCMP can adaptively choose the sampling order and combination according to the different costs, to achieve the optimal performance.

**Effects of Weight Parameter.** We show the cost and $\sqrt{\epsilon_{\text{PEHE}}}$ across $\lambda = 0.1, 0.3, 1, 3, 10$ on IHDP under $\alpha = 1$, *i.e.*, all-one cost in Figure 6. Similar results can be seen. These results reinforce the fact that we can tune the weight parameter $\lambda$ to the preference. On the other hand, policies almost degenerate at $\lambda = 0.1$ into giving action=0 directly, *i.e.*, stopping sampling directly before any observations are made. This shows that choosing an appropriate weight parameter is crucial to get the desired policy.

## B.2  ADDITIONAL POLICY VISUALIZATION

We visualize the same data in Figure 7. The same policies realize the same phenomenon. What's more, the large difference in our method compared to the time when alpha=1 is due to the different

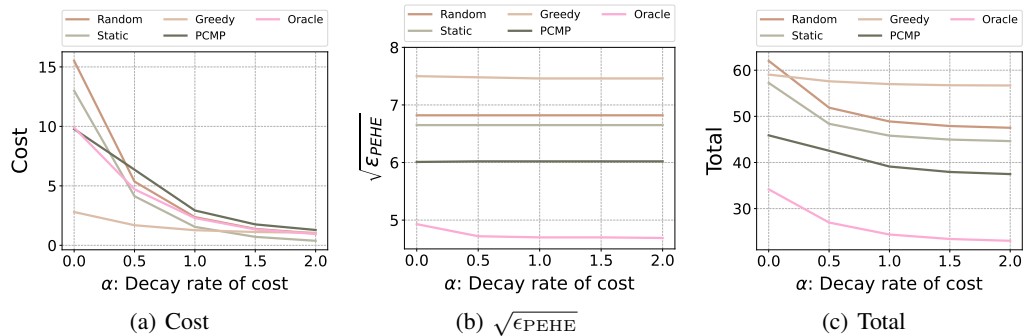

Figure 5: Comparative analysis of four baseline methods and the proposed PCMP on the IHDP. We leverage the causal method DESCN. We show the cost, $\sqrt{\epsilon_{\text{PEHE}}}$ and Total across $\alpha = 0, 0.5, 1.0, 1.5, 2.0$.

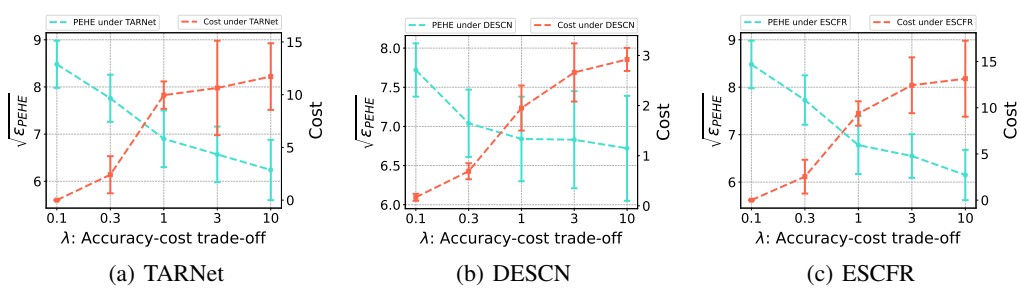

Figure 6: Comparative analysis of cost and $\sqrt{\epsilon_{\text{PEHE}}}$ across various weight parameter $\lambda$ on the IHDP under $\alpha = 0$, *i.e.*, all-one cost.

cost functions, and our method dynamically chooses different sampling strategies to achieve the lowest Total. This result highlights the benefits of the forward-looking and dynamic nature of our policy.

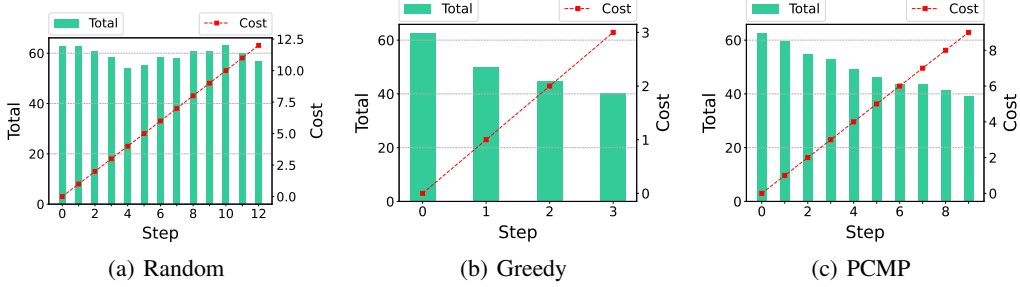

Figure 7: Visualizations of a test data from the IHDP dataset under $\alpha = 0$, *i.e.* all-one cost. For three policies, Random, Greedy, and PCMP, we show the accumulative costs and the Total across the acquisition steps.

### B.3 VISUALIZATIONS OF RL TRAINING

We randomly choose five datasets and show their training curve in Figure 8, Figure 9 and Figure 10. Notice that the sum of reward for most methods fluctuates upward and eventually converges gradually and smoothly. This reflects that our reinforcement learning algorithm learned great policies. However, there are still some strategies that suffer from crashes, *i.e.*, a sample length of 1 along with a sum

of the reward of 0, which suggests that the police always gives action directly to 0 thus terminating the sampling before any observation is made. At this point, we solve this problem by retraining by replacing the random seed or adjusting the initialization to learn a more reasonable policy for each dataset.

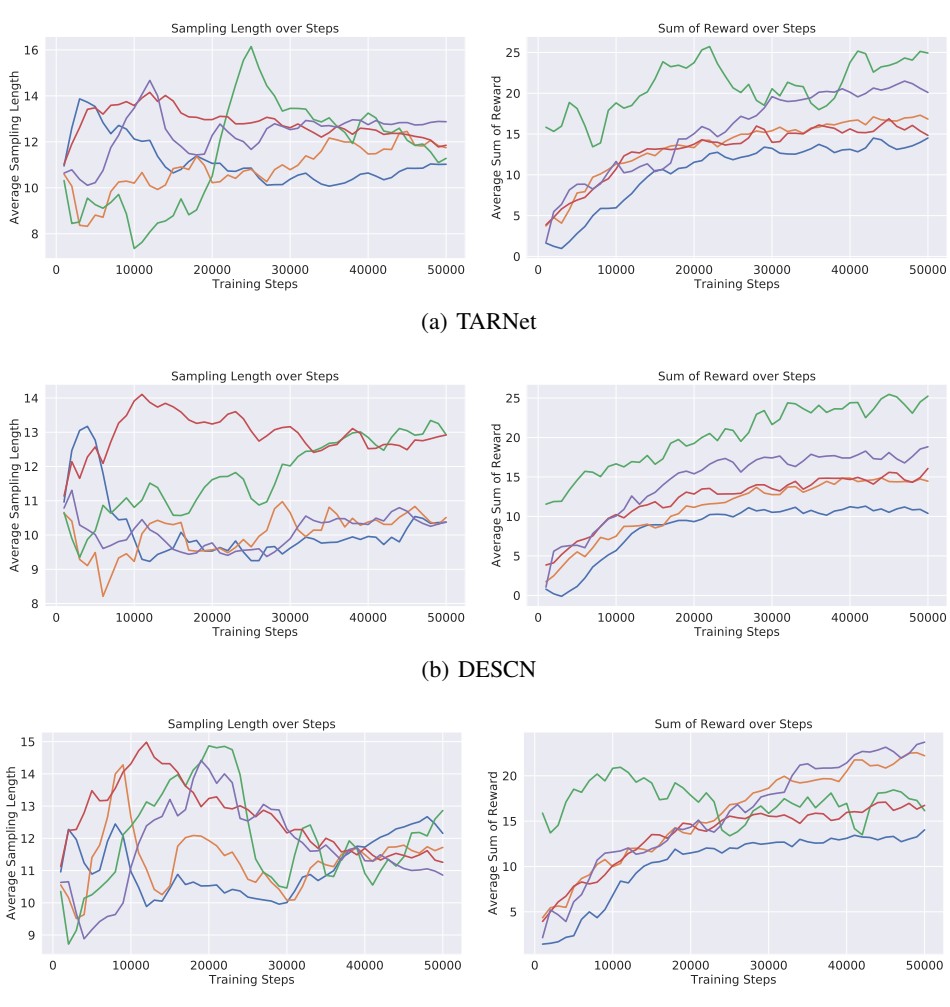

(a) TARNet

(b) DESCN

(c) ESCFR

Figure 8: Training curves of our policy on the IHDP dataset. We randomly select 5 datasets and plot the average sampling length over steps (left) and the average sum of reward over steps (right). The cost function is all-one cost and $\lambda = 1$. We set smooth to 0.6 for all curves.

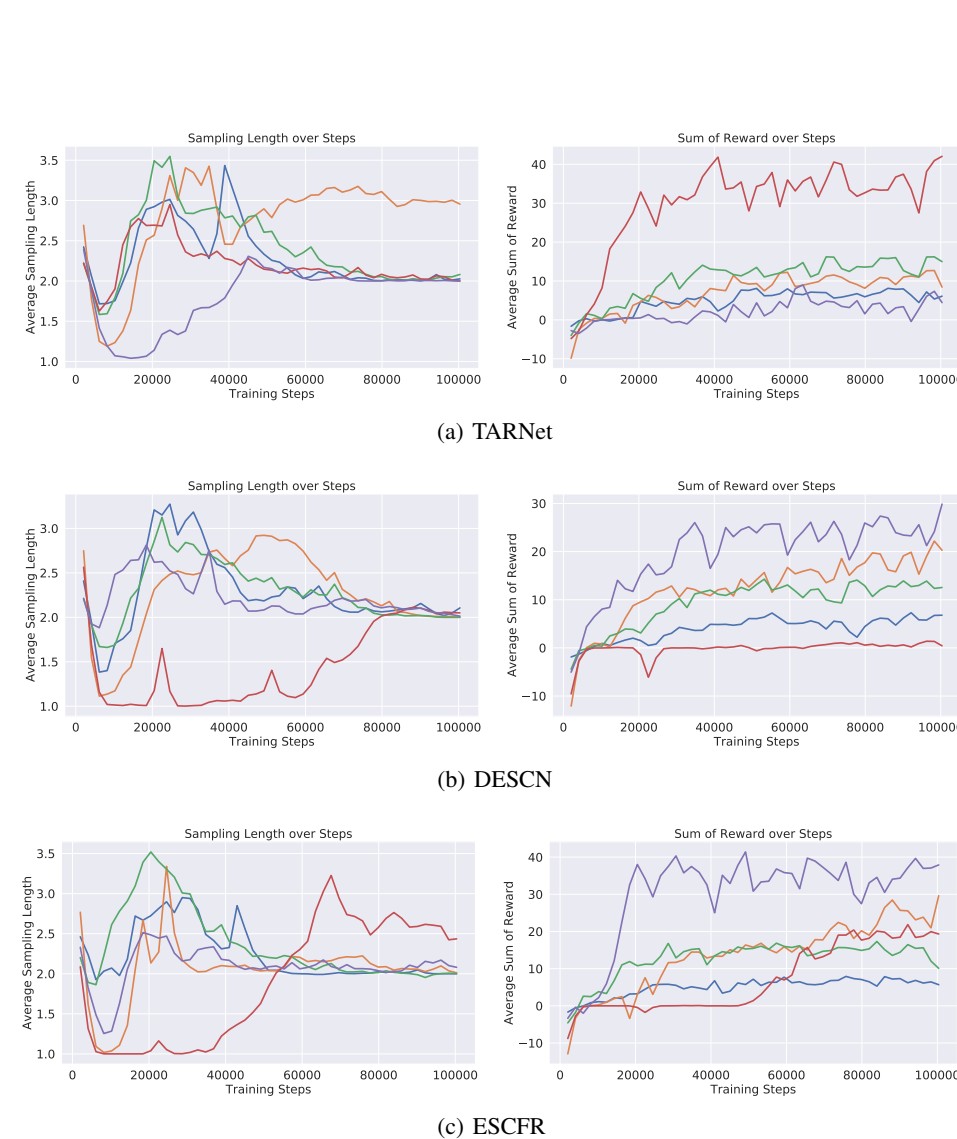

Figure 9: Training curves of our policy on the ACIC dataset. We randomly select 5 datasets and plot the average sampling length over steps (left) and the average sum of reward over steps (right). The cost function is all-one cost and $\lambda = 1$. We set smooth to 0.6 for all curves.

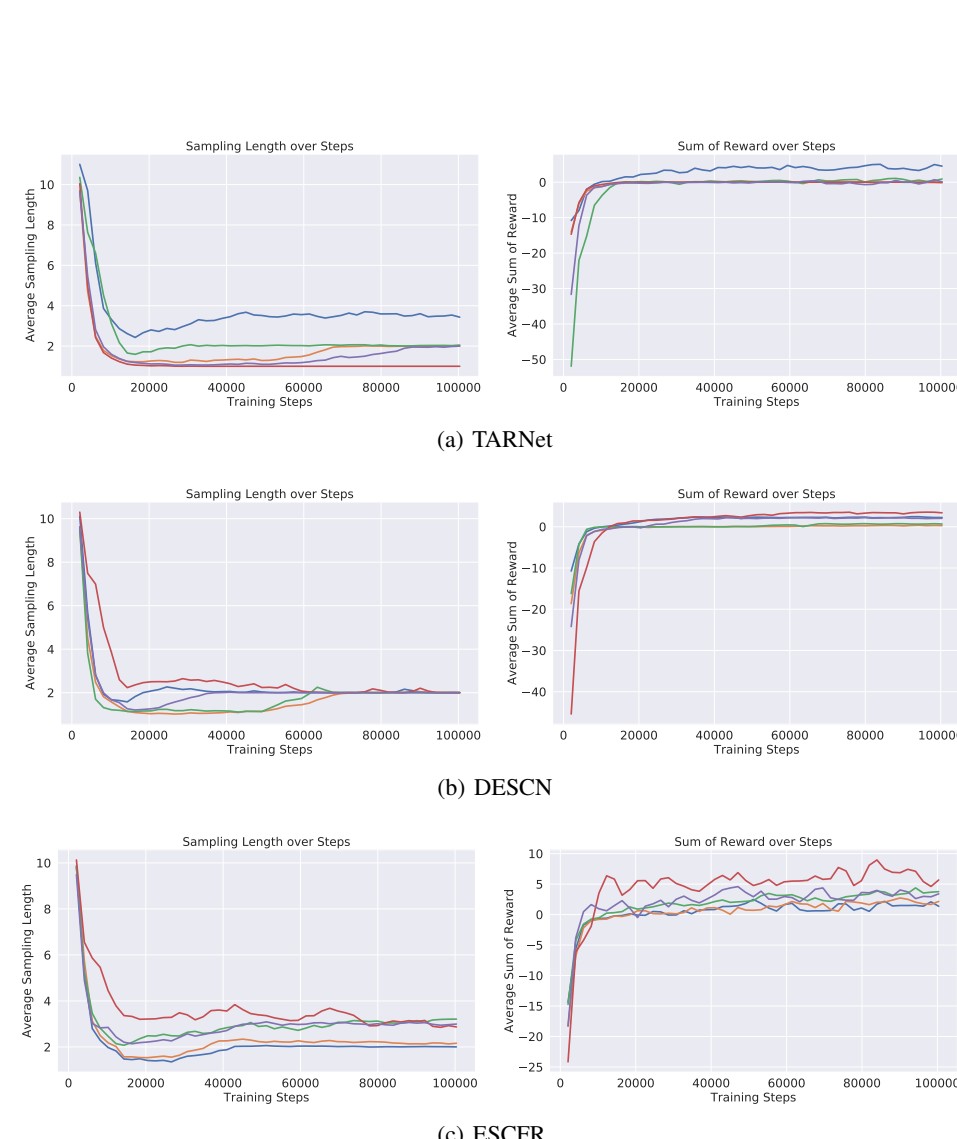

(a) TARNet

(b) DESCN

(c) ESCFR

Figure 10: Training curves of our policy on the Jobs dataset. We randomly select 5 datasets and plot the average sampling length over steps (left) and the average sum of reward over steps (right). The cost function is all-one cost and $\lambda = 100$. We set smooth to 0.6 for all curves.