# OpenReview forum: "Active Causal Learning for Conditional Average Treatment Effect Estimation"
_ICLR.cc/2025/Conference — Submitted to ICLR 2025_

### Official Review · Reviewer_gYcr · 2024-11-02

**Soundness:** 3
**Presentation:** 3
**Contribution:** 3
**Rating:** 6
**Confidence:** 3

**Summary:**

The paper proposes a new problem of covariate selection, aiming to balance effect estimation accuracy and covariate measurement cost. This paper proposes a Partially Observed Markov decision process (POMDP)-based method to learn the dynamic sampling strategy. The experimental results show the effectiveness of the proposed method.

**Strengths:**

1. The paper defines a novel covariate selection problem, which aims to balance effect estimation accuracy and covariate measurement cost.
1. The paper provides a POMDP-based solution, and the experiments verify its effectiveness.

**Weaknesses:**

1. For the problem formulation in Eq. (3), directly minimizing $\hat L_f$ might not be practical enough. For many applications, a hard constraint, i.e., $\hat L_f < \epsilon$ could be better.



Even though the proposed method might be simple, I still appreciate the proposed new setting and problem formulation.

**Questions:**

/

---

### Official Review · Reviewer_puau · 2024-11-03

**Soundness:** 2
**Presentation:** 2
**Contribution:** 2
**Rating:** 5
**Confidence:** 4

**Summary:**

This article describes how the covariates themselves are costly to collect, so the meaningful question is how to sift through the most informative covariates to guide causal learning at minimal cost.

**Strengths:**

Pros: the motivation for this article is relatively sound.

**Weaknesses:**

Cons: This article comes under the umbrella of active learning, and I am cautious that more causal active learning (at the theoretical level as well as at the applied level) needs to be extensively and deeply contrasted in the work related to this article; secondly, I think that the methodology of this article is a bit oversimplified and that the theoretical level of the analyses is a bit thin

**Questions:**

Could you give me more theoretical guarantee about the performance of your new method?

---

### Official Review · Reviewer_qo2N · 2024-11-03

**Soundness:** 1
**Presentation:** 2
**Contribution:** 1
**Rating:** 3
**Confidence:** 5

**Summary:**

The manuscript proposes a cost-efficient conditional average treatment effects (CATE) estimation method by sequentially querying covariates for a new subject. With partially observed covariates, the CATE estimator takes masked full covariates as the input to predict potential outcomes. Authors propose to learn a dynamic covariate-sampling strategy which can balance the cost of querying covariates and CATE loss. By formulating the optimization problem into a partially observable Markov decision process, authors leverage the PPO method to learn such sampling strategy.

**Strengths:**

Active querying in estimating causal effects is a interesting problem in the field. The article is well organized with extensive numerical studies.

**Weaknesses:**

- There is no identification for CATE with masked covariates. The SUVTA condition is only assumed for full covariates $X_i$. Please discuss how the SUTVA and other identification assumptions may need to be modified or extended when dealing with partially observed covariates. It is suggested to provide theoretical justification or empirical evidence for the validity of CATE estimation with masked covariates.

- The estimation model $f_w$ is trained only with fully observed $X$, i.e., the input is $f_w(\mathcal{X}(X,\mathbf{1}))$. For observation $\mathcal{X}(X,M)$ with other mask $M$, the performance of $f_w(\mathcal{X}(X,M))$ can be very poor. Have you considered training $f_w$ with various masked inputs to improve its performance on partially observed data? How might this affect the overall performance of the proposed method?

- It seems that the policy in (3) is a stochastic policy as authors propose to use PPO to learn this policy, there is a missing expectation over action $a_t$ according to $\pi$ in the objective function. Please clarify whether the policy is stochastic and, if so, explain how the expectation over actions is incorporated into the objective function. Also, why is there an indicator function in the constraints as $M$ is a binary mask vector?

- In the POMDP setting, the transition should be $\mathbb{P}: \mathcal{S} \times \mathcal{A}\rightarrow \mathcal{S}$, and reward does not depend on $\mathcal{O}$. The policy typically leverages the past history, not only the current observation. I suggest to discuss how incorporating past history into the policy might affect the performance of the proposed method.

- Only when the discount factor $\gamma=1$, the RL value function matches the objective in (3). Why choosing $\gamma < 1$ in the experiments? How this choice affects the relationship between the RL value function and the original objective?

**Questions:**

See weaknesses.

---

### Official Review · Reviewer_Dhna · 2024-11-04

**Soundness:** 2
**Presentation:** 3
**Contribution:** 2
**Rating:** 3
**Confidence:** 4

**Summary:**

Note: I have previously reviewed this paper for NeurIPS. The authors do not appear to have revised the paper, in particular regarding my earlier comments, so I have submitted the same review as before.

This paper considers a setting where the learner can sequentially decide which features to acquire for a given instance. A dataset with complete features is available, which is used to train a data acquisition policy with RL. The focus is on estimating conditional average treatment effects, which are assumed to be identified in the historical data and then estimated with an existing approach.

**Strengths:**

The idea of sequentially acquiring features is well motivated, as in sequential decisions about tests or diagnostics in medicine. The experimental results show improvements from using RL over reasonable baselines like greedy acquisition.

**Weaknesses:**

The idea of using RL to define a policy for sequential feature acquisition is not new. There is a great deal of work on this problem in the supervised setting. A few examples that come up from a quick search:

Li and Oliva. Active feature acquisition with generative surrogate models. ICML 2021.

Ghosh and Lan. DiFA: Differentiable Feature Acquisition. AAAI 2023.

Zheng Yu, Yikuan Li, Joseph Chahn Kim, Kaixuan Huang, Yuan Luo, Mengdi Wang Deep Reinforcement Learning for Cost-Effective Medical Diagnosis. ICLR 2023.

The paper doesn't discuss any of this related work or how the proposed methods/problem formulation is related. While this paper focuses on causal estimation, the way that it addresses the problem effectively reduces it to supervised learning: the predictions of an existing causal inference method using the historical (complete) data are used to provide surrogate outcomes, after which the proposed reward function treats it exactly as in the supervised case. I believe that the paper needs to either illustrate a fundamental difference between supervised learning and CATE estimation in this setting, or compare to existing methods in this literature and demonstrate improved performance.

**Questions:**

Can you comment on the relationship to this previous work?

---

### Meta-Review · Area_Chair_a96J · 2024-12-21

**Metareview:**

This paper proposes a dynamic sampling strategy to estimate conditional average treatment effects (CATE) from observational data while minimizing the cost of acquiring features and ensuring estimation accuracy.
The idea of combining the dynamic feature selection and CATE estimation is quite interesting and of practical importance; however, considering the lack of comparison with related work, insufficient theoretical guarantees, and limitations in experimental design, the paper does not meet the acceptance criteria at this stage.

**Additional Comments On Reviewer Discussion:**

Through the discussions, there were no significant changes in the scores.
Overall, while the novelty of the proposed method is appreciated, the lack of comparison with existing studies and theoretical guarantees kept the scores low.

---

### Decision · Program_Chairs · 2025-01-22

Reject